# Citizenship, Migration and Mobility in a Pandemic (CMMP): A global dataset of COVID-19 restrictions on human movement

**Lorenzo Piccoli**[1,2]*, **Jelena Dzankic**[1], **Didier Ruedin**[2,3]

**1** Robert Schuman Centre at the European University Institute, Florence, Italy, **2** University of Neuchâtel, Neuchâtel, Switzerland, **3** University of the Witwatersrand, Johannesburg, South Africa

* lorenzo.piccoli@eui.eu

## Abstract

This research note introduces a new global dataset, the Citizenship, Migration and Mobility in a Pandemic (CMMP). The dataset features systematic information on border closures and domestic lockdowns in response to the COVID-19 outbreak in 211 countries and territories worldwide from 1 March to 1 June 2020. It documents the evolution of the types and scope of international travel bans and exceptions to them, as well as internal measures including limitations of non-essential movement and curfews in 27 countries. CMMP can be used to study causes and effects of policy restrictions to migration and mobility during the COVID-19 pandemic. The dataset is available through Cadmus and will be regularly updated until the last pandemic-related restriction has been lifted or become long-term.

**Data Availability Statement:** The data is available through the CADMUS Research Data repository: https://cadmus.eui.eu/handle/1814/68358 and https://cadmus.eui.eu/handle/1814/68359.

## Introduction

Since the start of the COVID-19 pandemic, decisions aimed at reducing human movement across and within the states' borders have become a key element of governments' action. Systematic information on such limitations can help to explain why policy responses to the COVID-19 pandemic vary across countries and territories. It can also be used to assess which of these measures were more effective in slowing down/containing the spread of SARS-CoV-2, where, and why. This research note introduces the Citizenship, Migration and Mobility in a Pandemic (CMMP) dataset that tracks public decisions to regulate human movement in response to the COVID-19 outbreak in 211 countries and territories worldwide.

CMMP can be used to explore questions relevant for scholarship on migration and border studies, mobilities, citizenship, and policy diffusion. How did border restrictions evolve within and across states? What kind of exceptions do states grant, who are the target persons and why? Do restrictions on international mobility and domestic movement go hand in hand? Do we observe patterns of cooperation and fragmentation globally, or patterns that are replicated in multilevel European governance? Further to these questions, the dataset can also be used to study when and where particular restrictions were put in place. Combined with data such as the size of the country, human development index, regime type, the dataset can be used to

**Funding:** This research was supported by the National Center of Competence in Research nccr – on the move funded by the Swiss National Science Foundation (grant 51NF40-182897) and by the European University Institute (special grant on projects related to the Covid-19 pandemic).

**Competing interests:** The authors have declared that no competing interests exist.

discern the effects of the different kinds of restrictions on human movement on the diffusion of the virus.

While CMMP is not the only index of public health measures in response to COVID-19 [1–3], our data uniquely provide fine-grained information regarding the type of restrictions to human movement, the target of specific restrictions, as well as exceptions to them. We believe it is important to systematically study *in what ways* these measures change the international regulatory administrations that govern mobility, identify *what* groups were more strongly affected, and provide evidence of *whether* these restrictions were temporary and reversible. Taken together, this information can facilitate the work of a number of organizations (e.g., the International Civil Aviation Organization, International Organization for Migration, World Customs Organization, and World Health Organization) in drafting worldwide standards for international travel after the pandemic, thus minimizing the socio-economic impact of the global lockdown. These organizations, as well as governments, can vastly benefit from an evidence-based study of restrictions on human movement that were used to limit the spread of the pandemic.

## Data coverage, sources, and processing

CMMP brings together two sets of available information on the closure of national borders and restrictions to internal mobility. Data are collected from public sources: the repository of the International Organization for Migration (IOM), websites of governments and media. The first data cluster covers international travel restrictions in 211 countries and territories; the second covers restrictions to domestic mobility in 27 countries in Europe. Both data clusters start on 1 March 2020 and allow users to track and compare variation in governments' responses.

Information for both data clusters was collected by a team of twelve researchers and then centrally coded by the main investigators. The definition of the different forms of human movement and related restrictions was created deductively by the core team of researchers, based on the framing of the measures adopted by governments between January and March 2020. Systematic definitions are included in S1 and S2 Codebooks. For the first data cluster on international travel, which is wider in coverage and in depth, we subsequently asked four researchers who are not part of the core team to replicate the coding. The resulting categories resemble those by Hale et al. [1], where coverage between the two datasets overlaps. We also ran validity tests on a sample of five countries, which resulted in 94% correspondence. Details are included in S1 Validity test.

The dataset currently covers the first wave of public decisions to regulate human movement in response to the COVID-19, ending on 1 June 2020. We will continue updating the dataset as governments add new restrictions and roll back some of the measures. The second wave will expand the data both backwards and forwards, covering the period between 24 January 2020 and 24 October 2020. The dataset will thereafter be regularly updated until the last pandemic-related restriction has been lifted or become long-term (as per national legislation). The dataset is available open access through Cadmus.

### Data cluster 1: International travel restrictions in response to COVID-19

The first data cluster tracks the restrictions to international travel implemented by 211 countries and territories [4]. The indicators capture the type of restriction (no restriction, self-isolation, quarantine, screening, medical certificate, visa, no entry), the target (nationality/citizenship, presence, residence), and exceptions (family, government authorization, humanitarian workers, military, nationals, visa-type, official delegations, partner countries, residents, risk-free individuals, transit, transport personnel, work-related travel, other specific

exceptions) across countries and time. The types of restrictions are ranked by whether an individual is allowed to cross the border or not, capturing any further requirements before or after the border checkpoint. The full codebook is included in S1 Codebook. Simplified interactive visualizations of this data cluster can be accessed through the following link: https://tabsoft.co/3fxs9d0.

## Data cluster 2: Mobility and border control in response to COVID-19

The second data cluster correlates policies related to border control and policies regulating domestic mobility in 27 countries in Europe [5]. It provides information on the type of border control (open border, mostly open border, mostly closed border, closed border), and the policies related to human movement inside the states' borders (no restrictions; population invited to stay at home, schools closed; businesses and schools closed; curfews). The full codebook is included in S2 Codebook. The interactive visualization can be accessed through the following link: https://tabsoft.co/2YqC5is.

## Preliminary observations

### Data cluster 1: Evolution of international travel restrictions over time (by type)

In this section we advance some preliminary observations on the evolution of international travel restrictions (Fig 1), focusing on the expansion and retraction of different measures (self-isolation, quarantine, screening and testing, medical certificate, visa, no entry) between 1 March and 31 May 2020 (Fig 2).

**Self-isolation.** Self-isolation was first adopted in March 2020 by African and South American countries (e.g., Benin, Bermuda, Ecuador, and Paraguay). Under self-isolation measures individuals are allowed to enter the country in line with standard immigration policy and advised to isolate themselves for a pre-defined period of time and not to have social contacts. They may be required to fill out a form saying where they intend to stay, and authorities have the right to check on that address. Most countries subsequently abandoned self-isolation and moved towards a complete shutdown of the borders. In April 2020, 11 countries used self-

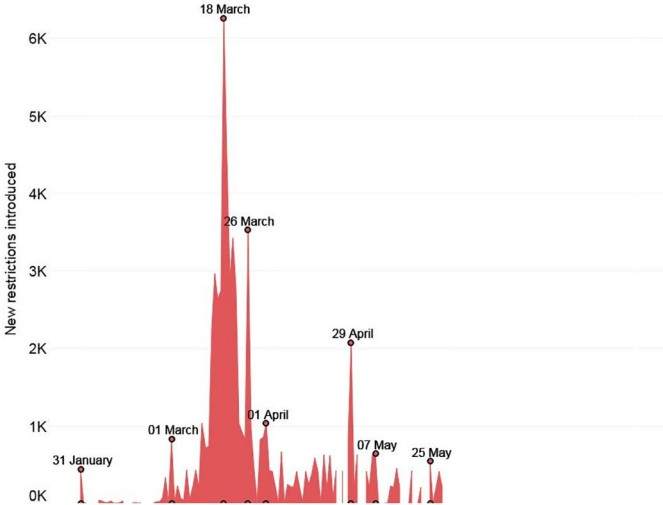

**Fig 1. Sum of all international travel restrictions introduced globally on a given day in the period between January and May 2020.** Source: International Travel Restrictions in Response to COVID-19 Dataset (2020).

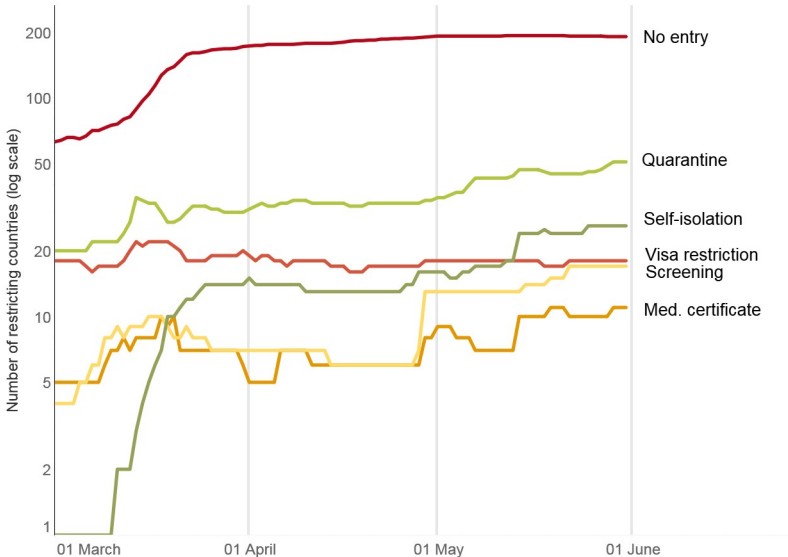

**Fig 2. Evolution of international travel restrictions over time (by type).** Source: International Travel Restrictions in Response to COVID-19 Dataset (2020).

isolation as the only tool to limit the spread of the virus from other countries. In May 2020, self-isolation was introduced to complement more restrictive 'no entry' measures in 14 other countries. In such cases, self-isolation applied to individuals otherwise exempt from travel bans.

**Quarantine.** Shortly after the World Health Organization (WHO) declared the COVID-19 pandemic on 11 March 2020, 25 countries introduced quarantine measures for international travellers. These measures allowed individuals to enter into the country in line with standard immigration policy but subjected them to mandatory quarantine in designated stations. With the increase of 'no entry' restrictions, 36 countries introduced the quarantine as a complementary measure, i.e., applied it to those who were exempt from the 'no entry' ban. Quarantine measures vary significantly in terms of the location and conditions of the facility (e.g., military barracks in Paraguay, five-star hotels in Cyprus), the length of isolation (e.g., 7 days in Slovenia, 21 days in the Central African Republic), and responsibilities for covering the costs (e.g., in Malaysia the government covers the costs for citizens but not for non-citizens).

**Screening and testing.** Under the mandatory screening measure individuals are allowed to enter the country in line with standard immigration policy if they are granted clearance by health control at the border. They may be refused entry or placed in mandatory quarantine —i.e., their movement restricted—if they fail the screening. Controls range from simple measurement of body temperature to extensive symptom checks and swab sampling. This measure was used by eight countries in March 2020. It sharply declined in use when COVID-19 became a global pandemic, also as the WHO was not supportive of this measure (World Health Organization, 2020, p. 24). In May 2020, 16 countries introduced mandatory antibody testing upon arrival for individuals exempt from the entry ban. Such individuals are admitted in the country, but subject to follow-up measures in case of a positive test (self-isolation, quarantine, etc).

**Medical certificate.** The requirement for individuals to provide a medical clearance ahead of travel was used by nine countries in the second half of March 2020. However, with the introduction of entry bans, medical certificate became obsolete. Throughout the months of

April and May 2020, however, ten countries re-introduced a mandatory prior medical certification for travelers exempt from the entry ban.

**Visa.** Twenty countries, mostly located in Asia and in the Middle East, suspended the issuing of all visas, or imposed pre-departure entry clearance that was not demanded before the outbreak of COVID-19. Eight countries abolished the visa requirement when they introduced the 'no entry' ban; 12 countries retained visa restrictions throughout the months of April and May 2020.

**No entry.** The first blanket prohibition to enter a country was introduced on 24 January 2020 by the Marshall Islands, which issued travel restrictions for all persons travelling via air or sea from the People's Republic of China [6]. No entry measures may apply selectively, to travelers from particular geographical locations, or they can limit all cross-border travel. The number of countries implementing a variation of a 'no entry' measure increased to 61 by 1 March 2020, 157 by 1 April, and 183 by 1 May. Our dataset shows regional diffusion patterns of this measure. In early March, most countries in Asia and in the Middle East introduced a 'no entry' ban. Between 13 March and 23 March 2020, there was a sharp increase in the use of the 'no entry' ban in Europe and the Americas. Travel restrictions, including closure of air and land borders, were put in place in late March 2020 in African countries.

## Target

In this section we discuss some preliminary observations on the evolution of international travel restrictions by target (presence, nationality, residence) between 1 March and 31 May 2020 (Fig 3).

**Presence.** The measure concerns travelers who had been physically present in one of the affected countries in the 14 days preceding the date of travel. These restrictions are, by far, the most widespread. Many governments adjusted their targets as time went by. In early February 2020, presence-related restrictions targeted individuals who had travelled to or transited viral hubs such as Lombardy region in Italy, or Wuhan in China. By mid-March, such travel bans extended to entire countries. In many cases, the list of targeted countries changed over time.

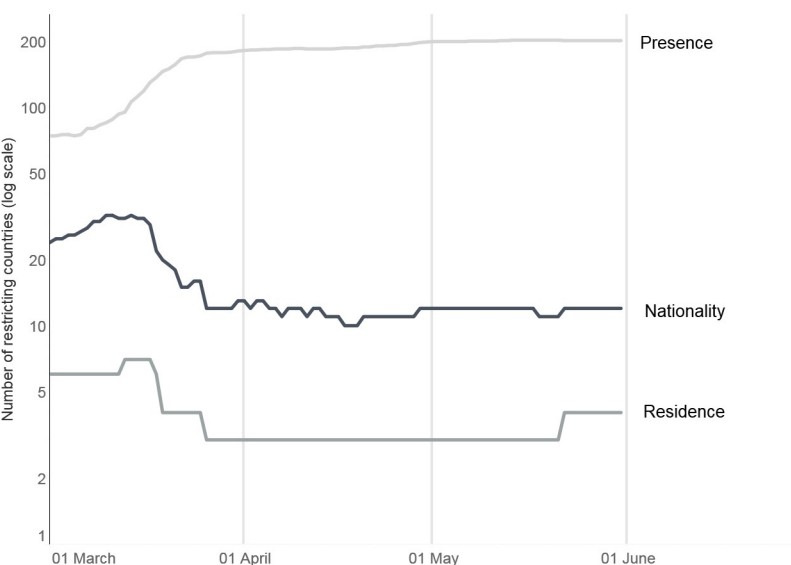

**Fig 3. Evolution of international travel restrictions over time (by target).** Source: International Travel Restrictions in Response to COVID-19 Dataset (2020).

For example, in the initial proclamation of 11 March 2020, the US travel ban applied to individuals coming from Schengen countries [7]. It was revised on 14 March to include also Ireland and the United Kingdom [8], while Brazil was added to the list in late May [9].

**Nationality.** In the early phases of the pandemic, twenty countries introduced 'no entry' restrictions based on nationality—a measure that may contravene article 12(4) of the International Covenant on Civil and Political Rights (ICCPR) guaranteeing readmission to nationals and settled residents–, applying to passport-holders of one of the affected countries. These measures also evolved over time. For instance, on 20 February 2020, the Russian government denied entry to Chinese nationals, unless in transit; on 28 February it banned the entry to all Iranian citizens; from 1 March South Korean nationals could only enter Russia via Sheremetyevo International Airport in Moscow; and from 13 March Russia denied entry to Italian citizens. A further example of a nationality ban is the case of Turkey, which denied entry to nationals of 69 countries regardless of their residence or previous stay. Overall, Chinese citizens have been the most frequent target of restrictions based on nationality. At the time of writing, Chinese with a passport issued in Hubei Province are still banned in Japan, South Korea, and Turkey.

**Residence.** Restrictions targeting travelers who legally reside in one of the affected countries was used by six countries in the early stages of the pandemic. Throughout April and May 2020, the governments of Hong Kong, Lithuania, and the Marshall Islands retained restrictions based on residence in China, Italy, Iran, and South Korea.

## Exceptions

Not everyone was affected equally by the travel restriction. Most states made exceptions for certain categories of people, allowing them to continue to cross their borders during the pandemic. In this section we present some preliminary observations on the evolution of these exceptions to international travel restrictions in the period between 1 March 2020 and 31 May 2020 (Fig 4).

**Citizens and residents.** Article 12(4) of the ICCPR forbids arbitrary deprivation of the right to re-enter one's own country. Most countries took a broad interpretation of this article, covering both the country of citizenship, and the country of residency in cases of permanent/long-term resident status. In fourteen countries, mainly in Asia and South America, only citizens were allowed to return. Some countries would not allow specific categories of residents to return. Indonesia and Japan denied re-admission to individuals without long-term residency

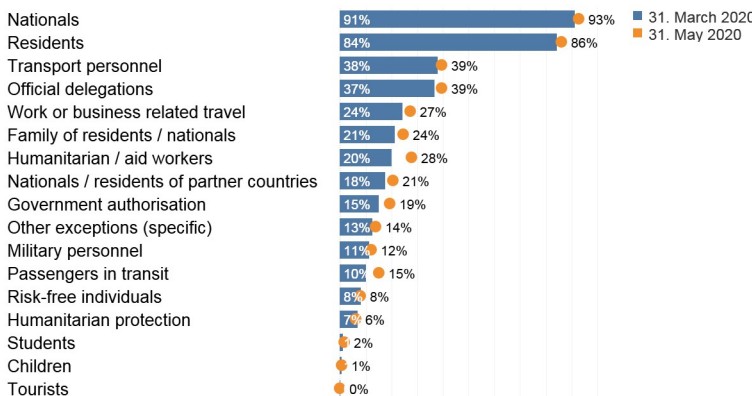

**Fig 4. Evolution of the exceptions to international travel restrictions over time.** Source: International Travel Restrictions in Response to COVID-19 Dataset (2020).

status, Mongolia would not guarantee re-entry to residents without personal ties with nationals, and Curacao to those without specific government authorization. While these and other decisions are highly idiosyncratic, depending on the type of residency permits available in different countries, we still capture them. In particular, while we subsume all residency-related restrictions in the same category, we provide more detailed information on the exact nature of each restriction in the interactive visualization available online.

By contrast, in some cases both citizens and residents were denied entry into the country. The Maltese government set a deadline on 12 April 2020 for Maltese nationals and residents to return [10], while China put a travel restriction on all foreign nationals, including those who have a valid residency permit, starting from 3 April 2020 [11].

**Family members.**   A total of 21 governments allowed family members of nationals and settled residents to return to the country. In May, the governments of Denmark, Germany, and France inserted new provisions to allow nonmarried couples to cross the borders, as long as they could provide proof of "a stable relationship".

**Transport personnel.**   In 39 countries goods transporters and commuters were exempt from travel bans. In the EU, they needed to show documentary proof of their right to cross the border.

**Other categories.**   Other exceptions include official delegations (54 exceptions), health care professionals and humanitarian workers (37 exceptions), international students (23 exceptions), individuals with a government authorization (21 exceptions), individuals with a temporary residence status (21 exceptions), and passengers in transit (20 exceptions). By contrast, asylum seekers (11 exceptions) and tourists (3 exceptions) were rarely allowed to cross international borders during the early phases of the pandemic.

In the Schengen Area, nationals of and residents in one of the 26 Schengen countries were frequently exempt from 'no entry' travel restrictions. Between early April and 1 July 2020, nationals of and residents in the Schengen area could transit through other countries to return to their country of nationality/residence.

Some countries allowed specific categories of workers (e.g., oil rig workers in Timor Leste), organized dedicated travel corridors (e.g., fruit pickers in the United Kingdom), or introduced exceptions to the closure of the borders (e.g., the border between Switzerland and Italy remained open to trans-border workers). In May 2020, only 12 countries granted an explicit exception for all individuals with a work authorization to cross the national border.

## Dataset 2: Policy convergence on closures and restrictions on internal mobility in Europe

In the EU, we observe an impressive policy convergence towards restrictive measures limiting human movement across the borders in the early stages of the pandemic. In particular, almost all EU countries closed the borders to non-essential travel in the second half of March 2020. However, severe restrictions to cross-border mobility were not automatically accompanied by restrictions to domestic mobility, such as lockdowns, curfews, etc. Some countries closed their borders but maintained internal mobility (e.g. Hungary, Estonia, Lithuania), while others shut down internal mobility without initially closing national borders (e.g. France, Italy, Spain).

By the end of April 2020, all 27 European countries covered in the dataset had introduced some limitations to internal mobility, although their restrictiveness varied significantly: curfew in eight states; closure of businesses and schools in eleven states; closure of schools only in eight states. Ireland and the United Kingdom stood out as the only countries that had not introduced restrictions to international mobility. Arguably, this was not necessary, since the volume of international travelling had naturally decreased by then.

Except for Switzerland, all other countries in the dataset took nation-wide decisions in the early phases of the pandemic. Only in early May 2020 did the governments of Germany and Italy adopt territorial-specific measures (e.g. additional restrictions to movement inside/outside the *regioni* and the *Länder*).

Finally, while in mid-May 2020 all 27 countries in the dataset started easing the restrictions to internal mobility, the borders remained mostly closed until the end of the month.

## Data limitations

CMMP has three limitations. Our first data cluster only records the date when border controls entered into force, and not the date when the measures were enacted. Dates of enactment and enforcement of measures may concur but adopted restrictions did not always have immediate effect. Given that CMMP focuses on restrictions on human movement, only the date when measures entered into force was captured. Diversity of legislative mechanisms (e.g., bill, executive decision, discretionary powers of presidency) also hampers a systematic capturing of enactment dates.

The second limitation of the dataset is that it documents legislative provisions, and not aspects related to their implementation. While Ireland, Mexico and the United Kingdom never technically ordered the closure of their borders, the extreme measures put in place at airports to stop the spread of the virus and travel restrictions ordered by their neighbors effectively cut them off most international travel.

Finally, CMMP does not register information on all restrictions on human movement. Specifically, it does not cover measures related to leaving one's country. Starting from 31 March 2020, for example, nationals of Qatar were not allowed to leave the country. In Singapore, the government also imposed stronger measures to discourage citizens from leaving—from 27 March 2020, those who traveled overseas and subsequently returned with COVID-19 would be charged non-subsidised rates for health treatment. These measures are relatively rare, but they are nonetheless important for a better understanding of how international movement was restricted during the COVID-19 pandemic.

## CMMP and future avenues of research

Restrictions to human movement both across and inside national borders can be conceived of as a global 'regime of mobility' [12], whereby states have the power and capacity to end international and internal mobility within days. The pervasive worldwide confinement reflects one of the core features of the modern state—its power to regulate human mobility [13]. CMMP shows that governments can introduce a broad range of measures to restrict human movement within and across their borders when necessary, but the nature and scope of these restrictions varies substantially. The spread of medical documentation and other requirements as conditions for border entry, for example, means that in many countries standard travel documents are no longer sufficient to cross national borders. At the same time, by capturing the wide range of exceptions granted by governments, CMMP also demonstrates that states are still dependent on human movement and therefore need to combine different tools to ensure the basic mobility.

The database can be used to address several key questions cutting across several disciplines. Most generally, CMMP provides groundwork for the evaluation of the impact of restrictions on various factors contributing to the transmission of the virus or its socioeconomic implications. More specifically, political and social scientists may use CMMP data to understand, explain, and compare the drivers of governments' choices. For mobility scholars, CMMP could be correlated to mobility patterns, testing the effect of stricter restrictions on actual

mobility. CMMP can also be a suitable tool for inter-country learning, policy diffusion, and mimicry across borders. It can show significant patterns of policy convergence and divergence. Additional attention could be devoted to study whether restrictions are discriminatory and, if so, against whom. Finally, information in the dataset can be analyzed from an epidemiological perspective to understand whether restrictions of human movement are effective in curbing the contagion. CMMP is well suited for this operation, since it allows for fine-grained testing as of the date of entry into force of a variety of measures, their targets, and the nature of restrictions, including exceptions.

## Supporting information

**S1 Codebook. International travel restrictions in response to the COVID-19 outbreak.** (DOCX)

**S2 Codebook. Mobility and border control in response to the COVID-19 outbreak.** (DOCX)

**S1 Validity test. International travel restrictions in response to the COVID-19 outbreak.** (XLSX)

## Author Contributions

**Conceptualization:** Lorenzo Piccoli, Jelena Dzankic.

**Data curation:** Lorenzo Piccoli, Jelena Dzankic, Didier Ruedin.

**Formal analysis:** Lorenzo Piccoli, Jelena Dzankic, Didier Ruedin.

**Investigation:** Lorenzo Piccoli, Jelena Dzankic.

**Methodology:** Lorenzo Piccoli, Jelena Dzankic.

**Software:** Didier Ruedin.

**Validation:** Lorenzo Piccoli, Jelena Dzankic.

**Writing – original draft:** Lorenzo Piccoli, Jelena Dzankic.

**Writing – review & editing:** Lorenzo Piccoli, Jelena Dzankic, Didier Ruedin.

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
