## [Decision Letter · Decision Letter 0]

24 Dec 2020

PONE-D-20-37625

Citizenship, Migration and Mobility in a Pandemic (CMMP): a global dataset of COVID-19 restrictions on human movement

PLOS ONE

Dear Dr. Piccoli,

Thank you for submitting your manuscript to PLOS ONE. After careful consideration, we feel that it has merit but does not fully meet PLOS ONE’s publication criteria as it currently stands. Therefore, we invite you to submit a revised version of the manuscript that addresses the points raised during the review process.

We look forward to receiving your revised manuscript.

Kind regards,

Francesco Di Gennaro

Academic Editor

PLOS ONE

Journal Requirements:

2.)  Please ensure that you refer to Figure 1, 2 and 3 in your text as, if accepted, production will need this reference to link the reader to the figure.

4.) Please include captions for your Supporting Information files at the end of your manuscript, and update any in-text citations to match accordingly. Please see our Supporting Information guidelines for more information: http://journals.plos.org/plosone/s/supporting-information

Additional Editor Comments:

dear authors follow reviewer suggestion to improve your excellent paper

Reviewers' comments:

Reviewer's Responses to Questions

**Comments to the Author**

1. Is the manuscript technically sound, and do the data support the conclusions?

Reviewer #1: Yes

Reviewer #2: Yes

Reviewer #3: Yes

2. Has the statistical analysis been performed appropriately and rigorously? 

Reviewer #1: Yes

Reviewer #2: Yes

Reviewer #3: N/A

3. Have the authors made all data underlying the findings in their manuscript fully available?

Reviewer #1: Yes

Reviewer #2: Yes

Reviewer #3: Yes

4. Is the manuscript presented in an intelligible fashion and written in standard English?

Reviewer #1: Yes

Reviewer #2: Yes

Reviewer #3: Yes

5. Review Comments to the Author

Reviewer #1: It is interesting study used for policy makers and scientist for managing the pandemic happening in the future. Also we can get insight from this finding how we reacted COVID 19. But I have minor concerns in Background section that the issues were not stated(what is the importance for conducting this study), the gap. Plus what is its impact and its implications? Great work!!!

. for the

Reviewer #2: The research work provides a rigorous review of national restrictions that will provide valuable groundwork for the evaluation of the impact of restrictions on various factors contributing to the transmission of the virus or its socioeconomic implications. Importantly, the database is updated regularly, provides the option to restrict evaluation to a limited time frame, and has initiated collection of domestic mobility restrictions that will be important for assessing the relationship between movement and transmission.

The methods are described clearly and the summary is straightforward, though lengthy. The methods section warrants a brief overview of the framework used to define and code the different forms of human movement and its restrictions.

The discussion of citizenship, however, is limited and the ICCPR is mentioned without initial definition or introduction. Within the exception section, what information on variances in residency type are collected if any? Within the target section, what data are collected on restrictions against asylum seekers, refugees, and other migrants which were central to many mobility restrictions and have separate international protections?

In dataset 2, it is stated that "we observe an impressive policy convergence towards restrictive measures limiting all forms of human movement." This is a blanket statement that without defining the spectrum of restrictions on human movement (with appropriate references) is nonfactual.

Overall an impressive repository, necessary dataset, and launchpad for critical future research.

Reviewer #3: The manuscript is not organized in as per the journal's instruction (title page, abstract, introduction, materials and methods, results, discussion, conclusions etc...); however, the nature of the data might affect them to do so.

6. PLOS authors have the option to publish the peer review history of their article (what does this mean?). If published, this will include your full peer review and any attached files.

Reviewer #1: **Yes: **Thomas Ayalew Abebe

Reviewer #2: No

Reviewer #3: No

---

## [Author Response · Author response to Decision Letter 0]

5 Feb 2021

We thank the reviewers for their careful comments. They helped us to better present the data and highlight their scientific relevance. Given the very detailed and straightforward feedback that we received, in what follows, we briefly respond how we have addressed each point individually.

Reviewer #1/1: What is the importance for conducting this study?

To further clarify the importance of the study, we have added a sentence on pp. 3-4: “We believe it is important to systematically study in what ways these measures change the international regulatory administrations that govern mobility, identify what groups were more strongly affected, and provide evidence of whether these restrictions are temporary and reversible.” In addition to this, we have amended the concluding section at pp. 17-18 to highlight many possible applications of the dataset and future research avenues.

Reviewer #1/2: What is the impact and implications?

On p.4, the text highlights the potential policy impact and implications of the dataset, especially for organizations such as the International Civil Aviation Organization, International Organization for Migration, World Customs Organization, and World Health Organization, but also national governments. Further to this, we have added a sentence on pp. 15-16: “In general terms, the dataset provides groundwork for the evaluation of the impact of restrictions on various factors contributing to the transmission of the virus or its socioeconomic implications”. Additionally, the concluding section at pp. 17-18 discusses many possible applications to explore the implications of the restrictions documented in the dataset. 

Reviewer #2/1: The methods section warrants a brief overview of the framework used to define and code the different forms of human movement and its restrictions.

We added an explanation on p. 4: “The definition of the different forms of human movement and related restrictions was created deductively by the core team of researchers, based on the framing of the measures adopted by governments between January and March 2020.”. Extensive documentation is provided with the dataset and in the Codebook (supplement S1). For this reason, we decided not to duplicate this information in the research note. We also mention the size of the research team to emphasize that the coding reflects an agreement by researchers from different disciplines. Finally, we note that the resulting categories resemble those by Hale et al. (2020) where coverage between the two datasets overlaps, providing external validity.

Reviewer #2/2: The discussion of citizenship, however, is limited and the ICCPR is mentioned without initial definition or introduction. 

We now introduce the ICCPR on p. 11. Given the focus of the research note, we have decided against a lengthy discussion of citizenship. On p. 5 we clarify that in the context of this research note, we use citizenship and nationality as synonymous.

Reviewer #2/3: Within the exception section, what information on variances in residency type are collected if any? 

We added the following explanation on p. 13: “While these and other decisions are highly idiosyncratic, depending on the type of residency permits available in different countries, we still capture them. In particular, while we subsume all residency-related restrictions in the same category, we provide more detailed information on the exact nature of each restriction in the interactive visualization available online”. At this stage, we do not code the exceptions, since they seem too specific to classify due to variations in the administrative definitions of residence across and within countries, but the description we include in the data will facilitate future coding efforts with specific research questions in mind.

Reviewer #2/4: Within the target section, what data are collected on restrictions against asylum seekers, refugees, and other migrants which were central to many mobility restrictions and have separate international protections?

We now addressed this comment on p. 14: “By contrast, asylum seekers (11 exceptions) and tourists (3 exceptions) were rarely allowed to cross international borders during the early phases of the pandemic”. Since we do not capture implementation, as noted in the conclusions, we unfortunately cannot elaborate about the actual practice for asylum seekers who often do not cross borders in a regular manner.

Reviewer #2/5: In dataset 2, it is stated that "we observe an impressive policy convergence towards restrictive measures limiting all forms of human movement." This is a blanket statement that without defining the spectrum of restrictions on human movement (with appropriate references) is nonfactual.

We thank the reviewer for spotting this: we have rephrased this sentence and added a qualifier on p. 15: “In the EU, we observe an impressive policy convergence towards restrictive measures limiting human movement across the borders in the early stages of the pandemic. In particular, almost all EU countries closed the borders to non-essential travel in the second half of March 2020”.

Reviewer #3:1: The manuscript is not organized in as per the journal's instruction (title page, abstract, introduction, materials and methods, results, discussion, conclusions etc...); however, the nature of the data might affect them to do so.

For this manuscript we have followed the structure that has become common practice in the field of index building in political sciences. Examples of widely cited notes of this kind include:

- Munck, G.L., and J. Verkuilen. 2002. Conceptualizing and Measuring Democracy. Comparative Political Studies 35(1): 5–34.

- Helbling, M., L. Bjerre, F. Römer, and M. Zobel. 2017. Measuring Immigration Policies: The IMPIC Database. European Political Science 16(1): 79–98.

- Schmid, S. D., Piccoli, L. and Arrighi, J.T. "Non-universal suffrage: measuring electoral inclusion in contemporary democracies." European political science 18.4 (2019): 695-713.

For the kind of article, we believe that this organization is more effective for readers, but we remain open to further feedback.

Other changes

We have also streamlined the text in several places. In particular, in order to keep the article focused on the presentation of the dataset, we have reworked concluding section on p. 17, presenting a sketch of future avenues of research. 

We have also added an additional figure (Figure 1) that displays the sum of all international travel restrictions introduced globally on a given day in the period between January and May 2020. This image is complementary to the discussion in the text and stands as an illustration of the evolution and frequency of restrictions to human movement across international borders.

---

## [Decision Letter · Decision Letter 1]

19 Feb 2021

Citizenship, migration and mobility in a pandemic (CMMP): A global dataset of COVID-19 restrictions on human movement

PONE-D-20-37625R1

Dear Dr. Piccoli,

We’re pleased to inform you that your manuscript has been judged scientifically suitable for publication and will be formally accepted for publication once it meets all outstanding technical requirements.

Kind regards,

Francesco Di Gennaro

Academic Editor

PLOS ONE

Additional Editor Comments (optional):

dear authors congratulations

Reviewers' comments:

Reviewer's Responses to Questions

**Comments to the Author**

1. If the authors have adequately addressed your comments raised in a previous round of review and you feel that this manuscript is now acceptable for publication, you may indicate that here to bypass the “Comments to the Author” section, enter your conflict of interest statement in the “Confidential to Editor” section, and submit your "Accept" recommendation.

Reviewer #2: All comments have been addressed

Reviewer #3: All comments have been addressed

2. Is the manuscript technically sound, and do the data support the conclusions?

Reviewer #2: (No Response)

Reviewer #3: Yes

3. Has the statistical analysis been performed appropriately and rigorously? 

Reviewer #2: (No Response)

Reviewer #3: N/A

4. Have the authors made all data underlying the findings in their manuscript fully available?

Reviewer #2: (No Response)

Reviewer #3: Yes

5. Is the manuscript presented in an intelligible fashion and written in standard English?

Reviewer #2: (No Response)

Reviewer #3: Yes

6. Review Comments to the Author

Reviewer #2: (No Response)

Reviewer #3: (No Response)

7. PLOS authors have the option to publish the peer review history of their article (what does this mean?). If published, this will include your full peer review and any attached files.

Reviewer #2: No

Reviewer #3: **Yes: **Yosef Gebreyohannes Abraha

---

## [Editor Report · Acceptance letter]

25 Feb 2021

PONE-D-20-37625R1 

Citizenship, migration and mobility in a pandemic (CMMP): A global dataset of COVID-19 restrictions on human movement 

Dear Dr. Piccoli:

I'm pleased to inform you that your manuscript has been deemed suitable for publication in PLOS ONE. Congratulations! Your manuscript is now with our production department. 

Kind regards, 

on behalf of

Dr. Francesco Di Gennaro 

Academic Editor

PLOS ONE